# First-Principles Investigation of Simultaneous Thermoelectric Power Generation and Active Cooling in a Bifunctional Semimetal ZrSeTe Janus Structure

**DOI:** 10.3390/nano14020234

**Published:** 2024-01-22

**Authors:** Brahim Marfoua, Jisang Hong

**Affiliations:** Department of Physics, Pukyong National University, Busan 48513, Republic of Korea; bmarfoua@pukyong.ac.kr

**Keywords:** ZrSeTe Janus, semimetal, anisotropy, thermoelectric, power generation, cooling performance

## Abstract

Traditional thermoelectric materials often face a trade-off between efficient power generation (high ZT) and cooling performance. Here, we explore the potential of achieving simultaneous thermoelectric power generation and cooling capability in the recently fabricated bulk ZrSeTe Janus structure using first-principles density functional theory (DFT). The layered ZrSeTe Janus structure exhibits a semimetal character with anisotropic transport properties along the in-plane and out-of-plane directions. Our DFT calculations, including the explicit calculation of relaxation time, reveal a maximum ZT of ~0.065 in the out-of-plane direction at 300 K which is one order of magnitude larger than that in the in-plane direction (ZT~0.006). Furthermore, the thermoelectric cooling performance is also investigated. The in-plane direction shows a cooling performance of 13 W/m·K and a coefficient of performance (COP_max_) of ~90 with a temperature difference (ΔT) of 30 K, while the out-of-plane direction has a cooling performance of 2.5 W/m·K and COP_max_ of ~2.5. Thus, the out-of-plane current from the thermoelectric power generation can be utilized as an in-plane current source for active heat pumping. Consequently, we propose that the semimetal ZrSeTe Janus structure can display bifunctional thermoelectric properties for simultaneous thermoelectric power generation and active cooling.

## 1. Introduction

Demand for sustainable energy solutions is growing exponentially, and this drives extensive research efforts on thermoelectric (TE) materials. Waste heat from various sources can be effectively converted into useful valuable electrical power using the TE material. Thus, it helps to minimize energy waste and environmental impact [1,2,3]. The performance of thermoelectric materials is commonly evaluated using the dimensionless figure-of-merit ZT defined by ZT = S^2^σT/κ. Here, S, σ, and κ stand for the Seebeck coefficient, electrical conductivity, and thermal conductivity (comprising electronic (*κ*_e_) and lattice (*κ*_L_) contributions), and T is the absolute temperature. Thus, a high-TE-performance material should have a larger power factor (S^2^σ) and lower heat dissipation (κ). On the other hand, thermoelectric cooling based on the Peltier effect also offers a unique approach to heat management. The cooling performance is quantified using the coefficient of performance COP = Q/W where Q represents the amount of heat transferred or delivered by the system, and W represents the work input or energy consumed by the system [4,5]. Thermoelectric cooling can be categorized into two modes: refrigeration mode and active cooling mode [5,6]. In the refrigeration mode, the Peltier current is flowing against the natural thermal flux resulting in heat (Q_C_) transfer from the cold side to the hot side which leads to a cooling effect on the cold side. In contrast, the Peltier current is reversed and it follows the natural thermal flux in the active cooling mode where the heat (Q_H_) is extracted from the hot side to the cold side. This mode can be employed in electronic cooling applications.

A high ZT is crucial for both thermoelectric power generation and thermoelectric refrigeration mode. For instance, commercially available thermoelectric Bi_2_Te_3_ alloys typically exhibit a ZT value of approximately 1 at around room temperature [5]. However, a typical COP of Bi_2_Te_3_ is around 0.7 [5,7], and this is much smaller than that in conventional refrigerators (COP~4–6). To achieve a COP of 4~6, the ZT should reach approximately up to 4 [7,8]. Unfortunately, such high ZT values in thermoelectric cooling systems are still far behind what can soon be achieved [5]. In contrast, ZT is not the sole relevant parameter [6,9,10,11] in the active cooling mode. In this active cooling mode, a combination of high active cooling (high power factor PF = S^2^σ) and high passive cooling (high thermal conductivity) determines the thermoelectric cooling performance. The thermal conductivity should be low for a high ZT but should be high for a high cooling efficiency. Generally, semiconductors may possess higher ZT values than metal systems, but a small cooling performance owing to the low thermal conductivity. In contrast, metals are superior to semiconductors in cooling performance due to their high thermal conductivity, but they lack thermoelectric power generation efficiency because of a low ZT. Thus, it is a very challenging task to find a material that possesses high thermoelectric power production (ZT) and an active cooling performance at the same time. In this aspect, semimetals may be positioned between semiconductors and conventional metals and offer a balanced performance. Therefore, we here aim to explore this challenging issue using a layered bulk semimetal material. Indeed, thermoelectric layered semimetals have recently emerged as potential candidates for TE-based device applications [12,13].

Layered semimetal systems may possess anisotropic electrical and thermal properties: high conduction along the in-plane direction, and suppression along the cross-plane direction for both electrons and phonons [14,15,16]. As a result, the out-of-plane direction can show a better thermoelectric power generation performance due to the larger Seebeck coefficient and lower thermal conductivity. Most semimetal systems have a ZT~0.1, but they still hold the potential for specific thermoelectric applications at ambient temperatures with strong directional anisotropy [17]. This advantage arises from their highly dispersive bands where the major charge carriers significantly outnumber the minor charge carriers [13]. On the other hand, the in-plane direction can have an enhanced thermoelectric cooling performance with a larger power factor and high thermal conductivity. Recently, a layered ZrSeTe Janus system has been synthesized using the chemical vapor transport (CVT) technique [18] and exhibits a transition from a semiconducting character in the ZrSe_2_ to a semimetal feature in the ZrSeTe Janus and ZrTe_2_. Many studies have been performed on the transport properties of the pristine ZrSe_2_ and ZrTe_2_ systems [19,20,21]. However, the thermal properties of the ZrSeTe Janus material have barely been investigated. Therefore, in this report, we aim to investigate the thermoelectric power generation and cooling performances of the layered ZrSeTe Janus system. Through systematic studies, we propose that the ZrSeTe Janus can be utilized for both thermoelectric power generation in the out-of-plane direction and a simultaneously enhanced thermoelectric cooling performance in the in-plane direction.

## 2. Numerical Method

### 2.1. Electronic, Phonon, and Lattice Thermal Conductivity Calculations

We employ the Vienna Ab-initio Simulation Package (VASP) and utilize the projector augmented wave (PAW) method [22,23,24]. The calculations are carried out using the generalized gradient approximation of Perdew–Burke–Ernzerhof (GGA-PBE) exchange-correlation functional [25]. Convergence criteria for the total energy and Hellman–Feynman forces are set to 10^−6^ eV and 0.001 eV/Å, respectively. A kinetic energy cutoff of 650 eV for the plane wave basis is used, and the Brillouin zone (BZ) is sampled with a k-mesh of 21 × 21 × 15 in the first Brillouin zone. To accurately account for the van der Waals (vdW) interactions in the layered system along the out-of-plane direction, we have incorporated vdW interactions based on the Grimme DFT-D3 method [26]. For the phonon and lattice thermal conductivity calculations, we use the Phonopy package to determine the second-order interatomic force constants (IFCs) using the finite-displacement method [27]. A supercell of 4 × 4 × 1 with k-point meshes of 3 × 3 × 1 is used to perform the phonon spectrum calculations. The lattice thermal conductivity is computed using the single-mode relaxation time approximation (RTA) and the linearized phonon Boltzmann equation implemented in the Phono3py code [28]. Third-order interatomic force constants were calculated using a 3 × 3 × 1 supercell with k-point meshes of 3 × 3 × 1. The lattice thermal conductivity is evaluated using a well-converged q-mesh in the reciprocal lattice space of 70 × 70 × 50.

### 2.2. Thermoelectric Properties Calculations

To calculate the thermoelectric coefficients, we use the BoltzTraP2 code [29] which utilizes the semi-classical Boltzmann transport theory. To ensure accurate Fourier interpolation of the Kohn–Sham eigenvalues, we increase the k-mesh density by 100 times in the irreducible Brillouin zone. The relaxation time is estimated using analytical energy-dependent expressions. Here, we consider various scattering mechanisms such as the fully anisotropic acoustic deformation potential, ionized impurity, and polar electron–phonon scattering. This approach provides a reasonable and computationally efficient approximation yielding comparable accuracy to the fully first principles approach. The relaxation time (τ) is defined as in [30]:(1)$$\tau \left(T,E\right)=\frac{1}{P_{imp}+P_{ac}+P_{polar}}$$
where the impurity scattering rate (P_imp_) is adopted from the Brooks–Herring formula $P_{imp}\left(T,E\right)=\frac{\pi n_{I}Z_{I}^{2}e^{4}E^{-3/2}}{\sqrt{2\bar{m}}{\left(4\pi \epsilon_{0}\epsilon_{s}\right)}^{2}}\left[\log\left(1+\frac{1}{x}\right)-\frac{1}{1\;+\;x}\right]$ with $x=\frac{\hbar^{2}q_{0}^{2}}{8\bar{m}E}$. The variables $n_{I}$, $Z_{I},\;\epsilon_{0,}{\;\epsilon }_{s}$, and $q_{0}\;$represent the ionized impurity concentration, impurity charge, vacuum permittivity, relative dielectric constant, and Debye screening wavevector. Acoustic phonon scattering is modeled within the deformation potential approach in the long wavelength acoustic-phonon limit $P_{ac}\left(T,E\right)=\frac{{\left(2\bar{m}\right)}^{3/2}k_{B}TD^{2}\sqrt{E}}{2\pi \hbar^{4}\rho v^{2}}$ where *E*, *D*, ρ, $\bar{m},\;$and v are the electron energy, deformation potential of the band energies calculated at the band extrema, mass density, average effective mass, and average sound velocity. For the optical polar scattering (P_polar_), we use the Ridley model [31] $P_{polar}\left(T,E\right)=\sum_{i}\frac{C\left(T,E,e_{i}^{LO}\right)-A\left(T,E,e_{i}^{LO}\right)-B\left(T,E,e_{i}^{LO}\right)}{Z\left(T,E,e_{i}^{LO}\right)E^{3/2}}$ where the sum is over all longitudinal-optical phonons with energy $e_{i}^{LO}$.

## 3. Results and Discussion

The 1T ZrSeTe Janus structure has a trigonal structure and space group P-3m1 (164) as demonstrated in the previous experimental report [18]. Figure 1a shows the schematic illustration of the top and side views. We obtain the lattice parameters of a = b = 3.80 Å and c = 6.35 Å after full geometry optimization. The in-plane lattice constants are close to the previous report on the 2D ZrSeTe Janus [32]. Note that the PBE-GGA functional was found to be in good agreement with the experimental measurement in the previous investigation of the ZrSeTe Janus [18]. Thus, we also calculate the electronic band structure using the PBE-GGA. Figure 1b shows the calculated band structure along the in-plane path (Γ-M-K-Γ) and the out-of-plane path (Γ-A) as indicated in the Brillouin zone (Figure 1c). The ZrSeTe Janus system has a semimetal band structure in both the in-plane and out-of-plane directions. In the in-plane direction, the Fermi energy crossing occurs near the Γ point in both Γ-M and K-Γ paths from the valence band, while the band crossing appears near the M point in the conduction band. In the out-of-plane direction, only the valence band has a Fermi energy crossing along the (Γ-A) path.

From the band dispersion, we expect that the out-of-plane transport may stem only from the valence band (hole doping) because the conduction band minimum (electron doping) has a very flat band. This implies an extremely large electron carrier effective mass, and consequently a very poor mobility. Therefore, we only consider the hole doped system in this report. In the in-plane direction, the band dispersions are similar along the Γ-M and K-Γ paths (xx and yy directions). This may indicate weak anisotropic electronic transport features in the in-plane directions. Since we aim to investigate the directional anisotropy between the in-plane and the out-of-plane directions, we consider only the K-Γ path in the in-plane direction. Note that the semi-classical Boltzmann transport theory implemented in the BoltzTraP2 code adopts the constant relaxation time (CRTA). However, it is crucial to include the energy-dependent relaxation effect. In this report, we calculate the energy-dependent hole carrier relaxation time based on the acoustic deformation potential, ionized impurity, and polar electron–phonon scattering using Equation (1). These scattering strengths can be extracted from the effective mass (m*), high-frequency ($\varepsilon_{\infty }$) and lattice ($\varepsilon_{L}$) dielectric constants, average sound velocity (v), deformation potential (*D*), the dominant longitudinal optical (LO) phonon frequency (ℏω_LO_), and the density of material (ρ). In Table 1, we present all these quantities of the ZrSeTe Janus system along the in-plane and out-of-plane directions with hole carrier charge.

Based on these, we estimate the total hole relaxation time at 300 K, 290 K, 280 K, and 270 K along the in-plane and out-of-plane directions. Figure 2a shows the calculated results. The hole relaxation time is on the scale of τ = 10^−14^ s, and this is on the same scale as that of conventional semimetals but shorter than that of semiconductors. The in-plane direction has a slightly longer relaxation time than the out-of-plane direction. Since the temperature range is rather narrow, no substantial temperature dependency is found at least within the range of 270~300 K. We further analyze the contributions to the relaxation from the acoustic (ac), polar electron–phonon (pol), and ionized impurity (imp) scatterings at 300 K. Figure 2b shows the calculated results. Here, we find that the polar electron–phonon scattering is the most dominant among the three scattering types. Note that the interactions between electrons and lattice vibrations (phonons) are on the order of a few meV. In addition, the ZrSeTe shows a vanishing energy gap, and its LO phonon frequency (ℏω_LO_) has energies of ~30.25 meV (as indicated in Table 1). This has led to a more efficient polar electron–phonon scattering process and dominant LO phonons in the ZrSeTe Janus system. This significant involvement of optical phonons in the conduction channel of the carriers results in the initial increase in the relaxation time. Subsequently, the relaxation time decreases up to the thermal activation energy level (k_B_T). The trend then reverses, with an eventual increase in total hole relaxation time attributed to the large doping effect.

Next, we calculate the electrical conductivity (*σ*). The electrical conductivity is relaxation-time dependent (*σ/τ*) as implemented in the BoltzTraP2 code and can be written as in [29]:(2)$$\sigma \left(T;\mu \right)=e^{2}\int \sigma_{\alpha \beta }(\varepsilon )\left[-\frac{\partial f_{\mu }(T,\;\;\varepsilon )}{\partial \varepsilon }\right]d\varepsilon$$
where *e* and *fμ* represent the electron charge and the Fermi–Dirac distribution function. *σ_αβ_*(*ε*) is the conductivity tensor which is defined as
(3)$$\sigma \left(\varepsilon ,T\right)=\int \sum_{b}v_{b,k}⨂{\;v}_{b,k}\tau_{b,k}\delta (\varepsilon -\varepsilon_{b,k})\frac{dk}{8\pi^{3}}$$
where *b* represents the Cartesian indices, *τ_b,k_* is the carrier-dependent relaxation time, and *v_b,k_* is the group velocity. Here, we calculate the electrical conductivity (*σ*) using the energy-dependent relaxation time (*τ*). Figure 3a shows the temperature-dependent electrical conductivity as a function of the hole carrier concentrations along the in-plane and out-of-plane directions. The magnitude of the electrical conductivity is on the scale of 10^6^ 1/Ω·m, and the maximum value is around 2.2 × 10^6^ 1/Ω·m at 300 K. Note that due to the small temperature range, the electrical conductivity is insensitive to the given temperature ranges. As expected in a layered system, the in-plane direction shows a larger magnitude of electrical conductivity than the out-of-plane direction. The extended electronic states within the layers allow a higher conductivity in the in-plane direction, and this causes directional anisotropy. A similar anisotropic behavior of the electrical conductivity was found in the pristine ZrTe_2_ system with a maximum value of ~6 × 10^6^ 1/Ω·m in the hole doping at 300 K [19]. This high electrical conductivity in the ZrTe_2_ may be attributed to the strong metallic character of the ZrTe_2_, while the ZrSeTe Janus has a semimetallic characteristic.

We now discuss the electronic part of the thermal conductivity (κ_e_). From the Boltzmann transport theory, the electronic thermal conductivity (κ_e_) can be written as in [29]:(4)$$\kappa_{e}(T,\;\mu )=\frac{1}{e^{2}T}\left(\int \sigma_{\alpha \beta }\left(\varepsilon \right){\left(\varepsilon -\mu \right)}^{2}\left[-\frac{\partial f_{\mu }\left(T;\varepsilon \right)}{\partial \varepsilon }\right]d\varepsilon \right)$$
where *σ_αβ_*(*ε*), *f_μ_*, *T*, *e*, *ε*, and *μ* represent the conductivity tensor, the Fermi–Dirac distribution function, the temperature, the elementary charge, the electron energy state, and the chemical potential. Note that the electronic thermal conductivity is relaxation-time dependent (*κ_e_/τ*) in the BoltzTraP2 code, and we apply the energy-dependent relaxation time (*τ*) to obtain the electronic thermal conductivity (*κ_e_*). Figure 3b shows the temperature-dependent electronic thermal conductivity as a function of the hole carrier concentrations along the in-plane and out-of-plane directions. The in-plane electronic thermal conductivity is almost 10 times larger than the out-of-plane direction, indicating a strong anisotropic behavior. The in-plane electronic thermal conductivity reaches ~15 W/m·K. The pristine ZrTe_2_ also showed similar anisotropic behavior (*κ*_e_~40 W/m·K with hole doping) at 300 K [19]. Note that both electrical conductivity (Figure 3a) and electronic thermal conductivity (Figure 3b) show a peak at the carrier concentration around 2 × 10^20^/cm^3^, which is inherently tied to the relaxation time that stems from the polar electron–phonon scattering features.

We now discuss the Seebeck coefficient. Unlike the electrical and electronic thermal conductivities, the Seebeck coefficient is carrier-relaxation-time independent in the framework of the semiclassical solution of the Boltzmann transport theory [29]. The Seebeck coefficient can be expressed as
(5)$$S(T,\;\mu )=\frac{1}{eT}\left(\frac{\int \sigma_{\alpha \beta }\left(\varepsilon \right)\left(\varepsilon -\mu \right)\left[-\frac{\partial f_{\mu }\left(T;\varepsilon \right)}{\partial \varepsilon }\right]d\varepsilon }{\int \sigma_{\alpha \beta }\left(\varepsilon \right)\left[-\frac{\partial f_{\mu }\left(T;\varepsilon \right)}{\partial \varepsilon }\right]d\varepsilon }\right)$$

Figure 3c shows the temperature-dependent Seebeck coefficient as a function of the hole carrier concentrations along the in-plane and out-of-plane directions. Unlike the electrical conductivity and the electronic thermal conductivity, the out-of-plane direction has a larger Seebeck coefficient than the in-plane direction. The maximum Seebeck coefficient is around 14 μV/K in the in-plane direction, while it is around 50 μV/K in the out-of-plane direction. This anisotropy arises from the anisotropic electron density near the Fermi level. The Seebeck coefficient displays a sign change with increasing hole doping concentration increases. The sign of the Seebeck coefficient along both directions depends on the electronic band structure indicating a change in the predominant charge carrier type from electron to hole concentrations. Note that the ZrSeTe Janus shows a larger Seebeck coefficient than the ZrTe_2_ metal system (~20 μV/K) at 300 K because of the alloying effect (Janus structure) [19].

Now we discuss the lattice thermal conductivity part (*κ_L_*) which can be expressed, based on the Boltzmann transport theory, as in [28]:(6)$$\kappa_{L}=\frac{1}{NV}\;\sum_{\lambda }K_{\lambda }=\frac{1}{NV}\;\sum_{\lambda }C_{\lambda }v_{\lambda }^{2}\tau_{\lambda }$$
where the *N* and *V* represent the unit cell in the system and its volume. The *C_λ_*, *v_λ_*, and *τ_λ_* are the mode heat capacity, phonon group velocity, and phonon lifetime.

The single-mode relaxation time *τ_λ_* is calculated from the phonon line width *Γ* of the phonon mode *λ* using the following relation [28]:(7)$$\tau_{\lambda }=\frac{1}{2\Gamma_{\lambda }(\omega_{\lambda })}$$
where *Γ* can be written as
(8)$$\Gamma =\frac{18\pi }{\hbar^{2}}\sum_{\lambda^{\prime }\lambda^{\prime \prime }\;}∆\;(-q+q^{\prime }+q^{\prime \prime })N_{\lambda^{\prime }\lambda^{\prime \prime }}(\omega ){\left|\Phi_{-\lambda \;\lambda^{\prime }\lambda^{\prime \prime }}\right|}^{2}\;$$

Here, *Φ*_-***λλ***′***λ***__″_ is the phonon–phonon interaction strength among the three phonons *λ*, *λ*′, and *λ*″, and this can be calculated from the second- and third-order force constant, while *N*_***λ***′***λ***__″_ (*ω*) is
(9)$$N_{\lambda^{\prime }\lambda^{\prime \prime }}\left(\omega \right)=\left(n_{\lambda^{\prime }}+n_{\lambda^{\prime \prime }}+1\right)\delta \left(\omega -\omega_{\lambda^{\prime }}-\omega_{\lambda^{\prime \prime }}\right)+\left(n_{\lambda^{\prime }}-n_{\lambda^{\prime \prime }}\right)\times \;\left[\delta \left(\omega +\omega_{\lambda^{\prime }}-\omega_{\lambda^{\prime \prime }}\right)-\;\;\;\;\delta \left(\omega -\omega_{\lambda^{\prime }}+\omega_{\lambda^{\prime \prime }}\right)\right]$$
with${\;n}_{\lambda }={\left[\exp\left(\frac{\hbar \omega_{\lambda }}{K_{B}T}\right)-1\;\right]}^{-1}.\;$ Figure 4a shows the temperature-dependent lattice thermal conductivity of the ZrSeTe Janus system. The in-plane lattice thermal conductivity is almost five times larger than the out-of-plane direction. For instance, the lattice thermal conductivity is around 4.2 W/m·K in the in-plane direction, while it is 0.68 W/m·K in the out-of-plane direction. In addition, the lattice thermal conductivity is always lower than the electronic thermal part, and this is typical behavior in semimetal and metal systems. The in-plane lattice thermal conductivity of the ZrSeTe Janus structure is three times smaller than that in the ZrSe_2_ pristine system (~12 W/m·K at 300 K) [33]. Figure 4b shows the phonon dispersion and partial density of states (PDOS). The acoustic phonons and low-frequency optical phonons are dominated by the Te atoms, while the Se and Zr atoms contribute to high-frequency optical phonons (above 4 THz). Figure 4c shows the phonon scattering rate. The acoustic phonons show a maximum scattering rate of ~0.10 ps^−1^, while the optical phonons exhibit a three-times-larger maximum scattering rate (~0.28 ps^−1^). This leads to a more prominent role of optical phonons in reducing thermal conductivity.

Combining all the transport coefficients, we calculate the thermoelectric power generation performance; figure of merit ZT = S^2^σT/κ. Figure 5a shows the temperature-dependent ZT along the in-plane and out-of-plane directions. Note that the ZT temperature dependence is too weak in the selected temperature range. The out-of-plane maximum ZT (ZT_max_) is larger than the in-plane ZT_max_, indicating a strong directional anisotropy. The difference is about one order of magnitude at the maximum ZT (ZT_max_). For instance, the out-of-plane ZT_max_ is ~0.065 with a carrier concentration of ~17 × 10^20^ hole/cm^3^, whereas the in-plane ZT_max_ is around 0.006 with a carrier concentration of ~1.5 × 10^20^ hole/cm^3^ at 300 K. We further analyze the ZT behavior based on the σ/*κ* ratio and S coefficient at a fixed temperature (300 K) and carrier concentration (17 × 10^20^ hole/cm^3^). The σ/*κ* ratio is almost the same in both the in-plane and out-of-plane directions. In contrast, the Seebeck coefficient (S) in the out-of-plane direction is five times larger than that in the in-plane direction. Hence, the directional anisotropy of the ZT is mainly governed by the Seebeck coefficient. For comparison, Figure 5b shows the temperature-dependent ZT_max_ of the ZrSeTe Janus system and other semimetal systems at 300 K [13,17,34,35]. The out-of-plane ZT of the ZrSeTe Janus is larger than that of other thermoelectric semimetal systems. Indeed, only the YbMnSb_2_ along the in-plane direction shows a larger ZT than the ZrSeTe Janus [13]. However, the ZrSeTe Janus has a better performance than the YbMnSb_2_ in the out-of-plane direction. Note that the growing of single crystals in the ZrSeTe Janus system is performed based on the chemical vapor transport methods along the out-of-plane direction. Thus, this ZT can be further improved using texture engineering [36].

We also investigate the thermoelectric cooling performance. The performance of thermoelectric cooling is defined as COP = Q_H_/W. From the heat current (Q_H_), the heat flux (Q′_H_) can be obtained where the maximum heat flux Q′_H-max_ is expressed as in [9]:(10)$${Q\prime }_{H-max}=(\frac{S^{2}\sigma T_{H}^{2}}{2∆T}+\kappa )\nabla T=(\frac{PFT_{H}^{2}}{2∆T}+\kappa )\nabla T$$

Here, *T_H_* is the hot side temperature, and ΔT is the temperature difference between the hot (*T_H_*) and cold (*T_C_*) sides. Note that S^2^σ represents the power factor (*PF*). Hence, the effective thermal conductivity (*κ**_eff_*) can be written as follows [9,37]:(11)$$\kappa_{eff}=\left(\frac{PFT_{H}^{2}}{2∆T}+\kappa \right)$$

Here, the term $\frac{PFT_{H}^{2}}{2∆T}$ represents the active cooling term, while κ is the passive cooling term. If both thermal conductivity and the power factor are large, then we can expect a high thermal performance. Then, the *COP_max_* in the cooling mode can be expressed as follows [9]:(12)$${COP}_{max}=\frac{T_{H}}{2T_{c}}+\frac{∆T}{ZT_{c}T_{H}}$$

Note that the *COP_max_* is strongly related to the temperature difference (Δ*T*) and unlike the thermoelectric refrigerator performance, the thermoelectric active cooler performs better under larger temperature differences as well as smaller *ZT* values. It is crucial to note that we here consider the *COP* under the dynamic process. In this dynamic regime, the heat dissipation and absorption at the cold and hot ends are not necessarily in equilibrium, where the system may not be under thermal equilibrium. Notably, we investigate the cooling performance and *COP_max_* at the carrier concentration which has *ZT_max_* in the out-of-plane direction. Figure 6a shows the cooling performance as a function of the temperature difference along the in-plane and out-of-plane directions. Here, the black and red lines refer to active and passive cooling, while the blue line represents the overall performance. Note that the hot side temperature is fixed at 300 K in the rest of our calculations. The in-plane direction shows a much larger cooling performance than the out-of-plane direction. For instance, the in-plane cooling performance is around ~13 W/m·K, while the out-of-plane direction has a cooling performance of ~2.5 W/m·K with ΔT of 30 K. This difference is mainly due to the larger passive cooling (thermal conductivity (*κ*)) in the in-plane direction, although the active cooling is larger along the out-of-plane direction owing to the larger out-of-plane Seebeck coefficient. In the in-plane direction, the passive cooling is always much larger than the active cooling. On the other hand, both active and passive cooling are comparable in the out-of-plane direction. The active cooling is larger up to the ~Δ*T* of 10 K. After that, the passive cooling process dominates. We also estimate the COP_max_ along the in-plane and out-of-plane directions. Figure 6b shows the calculated COP_max_ results. The in-plane *COP_max_* is about 90 with Δ*T* = 30 K, and this is much higher than the out-of-plane *COP_max_* (~2.5 with Δ*T* = 30 K). The *COP_max_* is inversely proportional to the *ZT*, and the in-plane direction has a much smaller ZT than the out-of-plane direction. Therefore, the in-plane *COP_max_* performance is better than the out-of-plane. While the thermoelectric cooling performances of the copper (*κ**_eff_*~350 W/m·K with Δ*T* of 50 K at 500 K) [38] are far ahead, the ZrSeTe Janus system shows an almost three-times-larger COP than the CePd_3_ near room temperature (*κ**_eff_* ~ 33 W/m·K with Δ*T* of 30 K at 350 K) [38]. Indeed, most well-known thermoelectric cooling systems have a poor thermoelectric power generation performance (low *ZT*) owing to their strong metallic character. Therefore, it is rare to find a material that displays both thermoelectric power generation and thermoelectric cooling in one system for electronic cooler systems. These results may suggest that the ZrSeTe Janus system possesses bifunctional thermoelectric features. For instance, the out-of-plane direction can efficiently convert waste heat into electrical current through thermoelectric power generation. This generated electrical current can then be utilized in the in-plane direction to actively pump heat from the hot side to the cold side, leveraging the thermoelectric cooling effect as shown in Figure 6c. By combining these functionalities, the ZrSeTe Janus may possess the potential to act as a self-powered cooler for cooling hot electronic components.

## 4. Conclusions

In summary, we investigate the thermal transport properties of the layered ZrSeTe Janus structure. The ZrSeTe Janus structure shows a semimetallic band structure. The in-plane direction has a higher electrical conductivity and electronic thermal conductivity than the out-of-plane direction, while the out-of-plane direction shows a larger Seebeck coefficient. The lattice thermal conductivity is significantly lower than the electronic thermal conductivity due to its semimetal characteristics. Moreover, the lattice thermal conductivity shows anisotropic behavior. For instance, the in-plane direction has a higher value than the out-of-plane direction. The figure of merit *ZT* also displays anisotropic behavior. The out-of-plane direction *ZT* is ~0.065 at 300 K, while the in-plane direction *ZT* is one order of magnitude smaller. We also investigate the coefficient of performance (*COP*) based on effective thermal conductivity (*κ**_eff_*). The in-plane direction exhibits a significantly higher cooling performance than the out-of-plane direction due to the larger passive cooling (thermal conductivity) in the in-plane direction. The in-plane cooling performance is around *κ**_eff_*~13 W/m·K and *COP_max_*~90 with ΔT = 30 K, while the out-of-plane direction shows a cooling performance of *κ**_eff_* ~2.5 W/m·K and *COP_max_* ~ 2.5 with Δ*T* of 30 K at the room temperature hot side. As a result, the out-of-plane current from the thermoelectric power generation can be utilized as an in-plane current source for active heat pumping. It is worth noting that semiconductors possess a high thermoelectric power generation efficiency due to their large *ZT* values but their low thermal conductivity hampers potential thermoelectric cooling applications. On the other hand, metals are usually superior to semiconductors in cooling due to their high thermal conductivity but lack power generation efficiency due to the low Seebeck coefficient and large thermal conductivity. Therefore, semimetals may offer balanced performances in both thermoelectric power generation and thermoelectric cooling performances. Notably, we propose that the ZrSeTe Janus structure can display bifunctional thermoelectric properties for simultaneous power generation and active cooling usage.

## Figures and Tables

**Figure 1 nanomaterials-14-00234-f001:**
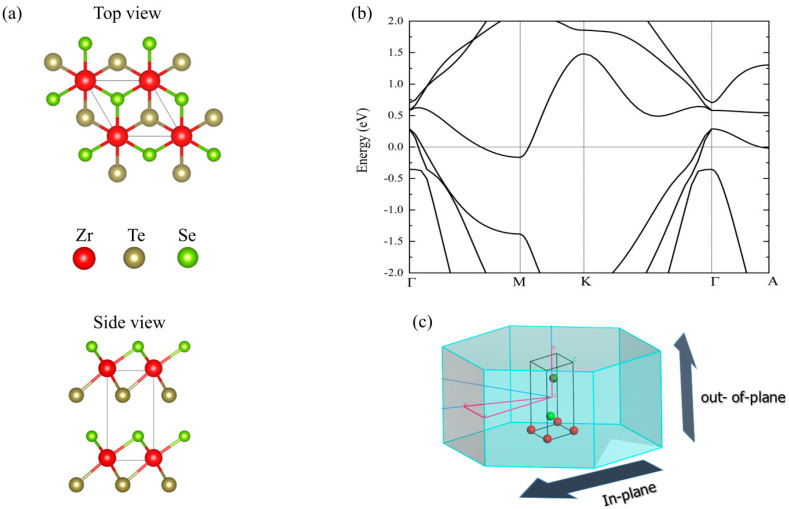
(**a**) Top and side views of the ZrSeTe Janus structure. (**b**) Electronic band structure of ZrSeTe Janus. (**c**) Illustration of the in-plane and out-of-plane direction along the high symmetric points in the first Brillouin zone (BZ).

**Figure 2 nanomaterials-14-00234-f002:**
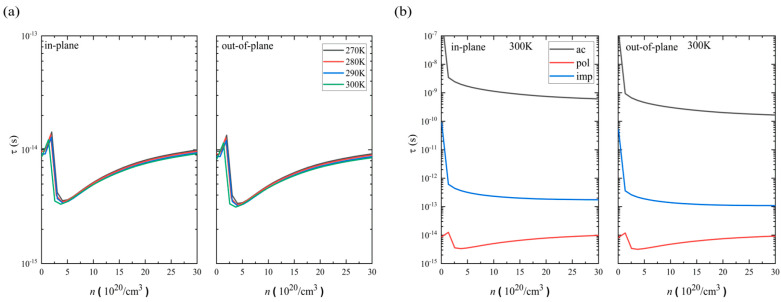
(**a**) Temperature-dependent hole carrier relaxation time (τ_tot_) in the in-plane and out-of-plane directions. (**b**) Contributions to the relaxation time from impurity (imp), acoustic phonon (ac), and optical polar phonon (pol) scatterings at 300 K of ZrSeTe Janus.

**Figure 3 nanomaterials-14-00234-f003:**
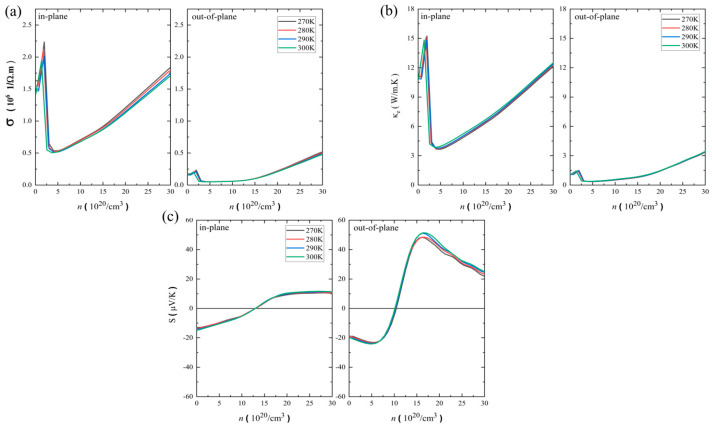
Temperature-dependent (**a**) electrical conductivity, (**b**) electronic thermal conductivity, and (**c**) Seebeck coefficient in the in-plane and out-of-plane directions of ZrSeTe Janus.

**Figure 4 nanomaterials-14-00234-f004:**
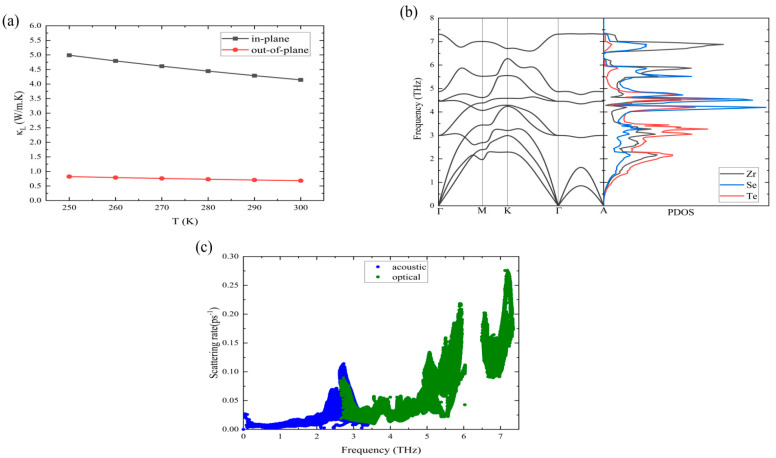
(**a**) Temperature-dependent lattice thermal conductivity (κ_L_) in the in-plane and out-of-plane directions of ZrSeTe Janus. (**b**) Phonon dispersion and partial density of states (PDOS) of ZrSeTe Janus. (**c**) Acoustic and optical phonon scattering rate of ZrSeTe Janus.

**Figure 5 nanomaterials-14-00234-f005:**
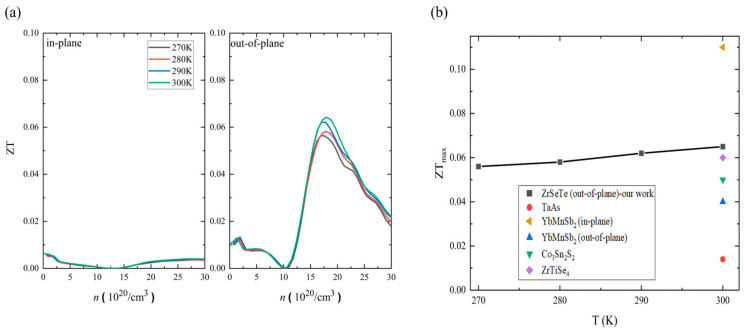
(**a**) Temperature-dependent ZT in the in-plane and out-of-plane directions of ZrSeTe Janus. (**b**) Temperature-dependent ZT maximum (ZT_max_) of ZrSeTe Janus in comparison with the ZT_max_ of other thermoelectric semimetals at 300 K.

**Figure 6 nanomaterials-14-00234-f006:**
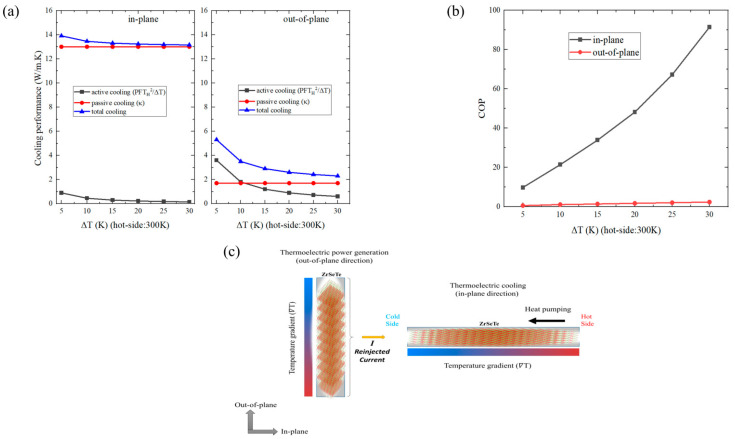
(**a**) Cooling performance as a function of the temperature difference (Δ*T*) in the in-plane and out-of-plane directions of ZrSeTe Janus. Here, the black and red lines refer to the active cooling and passive cooling while the blue lines are for the total cooling performance. (**b**) The coefficient of performance (COP_max_) as a function of the temperature difference (Δ*T*) in the in-plane and out-of-plane directions of ZrSeTe Janus. The hot side is fixed at 300 K. (**c**) Schematic illustration of the bifunctional thermoelectric properties in the ZrSeTe Janus structure along the in-plane (thermoelectric cooling) and out-of-plane (thermoelectric power generation) directions.

**Table 1 nanomaterials-14-00234-t001:** The effective mass ratio (m*/m), DP constant (D), high-frequency ($\varepsilon_{\infty }$) and lattice ($\varepsilon_{L}$) dielectric constants, average sound velocity (v), the dominant LO-phonon frequency (ℏω_LO_), and the density of material (ρ).

System	Direction	Carrier Type	m*/m	*D* (eV)	$$\varepsilon_{\infty }$$	$$\varepsilon_{L}$$	v (km/s)	ℏω_LO_ (meV)	ρ (kg/m^3^)
ZrSeTe	in-plane	hole	0.14	1.64	2.11	1.00	9.05	30.25	6217
	out-of-plane	hole	0.2	0.59	1.49	1.00	2.21	30.25	6217

## Data Availability

The raw data that support the findings of this study are available from the corresponding author upon reasonable request.

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
