# Peer review of "First-Principles Investigation of Simultaneous Thermoelectric Power Generation and Active Cooling in a Bifunctional Semimetal ZrSeTe Janus Structure"

_nanomaterials, 2024, doi:10.3390/nano14020234_

Round 1

Reviewer 1 Report

Comments and Suggestions for Authors

Semimetals are potential thermoelectric materials due to the unique electronic band structure. In this work by Marfoua et al., both the in-plane and out-of-plane electrical and thermal properties of the ZrSeTe semimetal have been calculated by first-principles calculations and semi-classical Boltzmann transport theory. The anisotropic feature of ZrSeTe results in higher dimensionless thermoelectric figure of merit along the out-of-plane direction as well as better active cooling performance in the in-plane direction. Overall, the research results are well presented in the paper, and the content is interesting to a wide broad readership. The work might be accepted after an appropriate revision.

1.      The relaxation time (Figure 2), electrical conductivity (Figure 3a) and electronic thermal conductivity (Figure 3b) show a peak at the carrier concentration around 2×10^20 cm^-3. Could the author explain this?

2.      The ZrSeTe semimetal shows very trivial thermoelectric performance, for example, the maximum zT value is only ~0.065 at 300K in the out-of-plane direction. Why do the authors think that the ZrSeTe semimetal can display bifunctional thermoelectric properties for simultaneous thermoelectric power generation and active cooling?

3.      There are still some spelling mistakes in the text. For example, in Table 1, Km/s should be km/s, and Kg/m^3 should be kg/m^3. The mass density of ZrSeTe can not be as large as 117231 kg/m^3 (see Table 1).

Comments on the Quality of English Language

There are still some spelling mistakes in the text as mentioned in my comments to the author.

Author Response

ANSWERS TO REVIEWERS

We thank the reviewers very much for reading our manuscript carefully and providing valuable comments and suggestions, which help us to enhance the quality of the work. That will greatly improve the manuscript and we have tried to do our best to respond to the points raised.

Reviewers have brought up worthy points and we appreciate the opportunity to clarify our research objectives and results. We have provided a detailed point-to-point response to address the following concerns. We have made necessary changes accordingly to their indications.

Our response follows.

Reviewer #1:

Semimetals are potential thermoelectric materials due to the unique electronic band structure. In this work by Marfoua et al., both the in-plane and out-of-plane electrical and thermal properties of the ZrSeTe semimetal have been calculated by first-principles calculations and semi-classical Boltzmann transport theory. The anisotropic feature of ZrSeTe results in higher dimensionless thermoelectric figure of merit along the out-of-plane direction as well as better active cooling performance in the in-plane direction. Overall, the research results are well presented in the paper, and the content is interesting to a wide broad readership. The work might be accepted after an appropriate revision.

C1.      The relaxation time (Figure 2), electrical conductivity (Figure 3a) and electronic thermal conductivity (Figure 3b) show a peak at the carrier concentration around 2×10^20 cm^-3. Could the author explain this?

Ans1: We thank the reviewer for this valuable comment and give us the opportunity to clarify this point. We attribute these behaviors to the intricate interplay of relaxation time dynamics and scattering mechanisms.

First, both electrical conductivity and electronic thermal conductivity are inherently tied to the relaxation time in the Boltzmann transport equation. Consequently, their behaviors strongly correlated to the temporal aspects of relaxation time.

In our work, we analyze the contributions of different scattering mechanisms to the relaxation time; acoustic phonon (ac), polar electron-phonon (pol), and ionized impurity (imp) scatterings at 300 K (depicted in Fig. 2(b)). We found that the polar (optical phonon) electron-phonon scattering is dominant among these three scattering mechanisms. Specifically, the polar electron-phonon scattering has a peak at the aforementioned carrier concentration (2×1020 cm-3).

In typical semiconductors, the relaxation time initially decreases with increasing carrier concentration (non-degenerate regime), reaching a minimum at the thermal activation energy level (kBT). Then, the relaxation time begins to rise as carrier concentration increases (the degenerate regime). However, our system has a semi-metallic band structure. Due to its gapless band structure, optical phonons can be involved in the conduction channel of the carriers. This distinctive feature leads to an initial increase in relaxation time with rising carrier concentration until the relaxation time reaches that specific carrier concentration corresponding to the optical phonon energy. After that, the relaxation time decreases up to the carrier concentration equivalent to the thermal activation energy level (kBT). In the end, the relaxation time increases again due to the large doping effect.

Overall, the peaks in relaxation time, electrical conductivity, and electronic thermal conductivity at a carrier concentration of 2×1020 cm-3 stem from the polar electron-phonon scattering coupled with the unique characteristics of the semimetal band structure. Then, the interplay of optical phonon frequencies and thermal activation energy results in a non-trivial trend in relaxation time.

Note that further discussion about this point is added in the revised manuscript.

C2.      The ZrSeTe semimetal shows very trivial thermoelectric performance, for example, the maximum zT value is only ~0.065 at 300K in the out-of-plane direction. Why do the authors think that the ZrSeTe semimetal can display bifunctional thermoelectric properties for simultaneous thermoelectric power generation and active cooling?

Ans2: We thank the reviewer for this valuable comment and give us the opportunity to clarify this point. Indeed, as commented, the ZrSeTe semimetal shows trivial ZT performance in comparison with typical thermoelectric semiconductor performance. Nevertheless, the thermoelectric potential of semimetals may offer advantages for specific low-power applications at room or low temperatures, such as powering wearable electronic devices or remote sensors requiring minimal power consumption 1. On the other hand, thermoelectric semimetals may be favorable for cooling applications despite their potentially lower ZT values. While high ZT thermoelectric semiconductors often have low thermal conductivity which limits their thermoelectric cooling applications, thermoelectric semimetals offer unique advantages for cooling applications due to higher thermal conductivity compared to thermoelectric semiconductors. In light of this, our study proposes the integration of thermoelectric power generation and cooling systems by utilizing the out-of-plane direction for thermoelectric power generation and the in-plane direction for enhanced thermoelectric cooling performance.

C3.      There are still some spelling mistakes in the text. For example, in Table 1, Km/s should be km/s, and Kg/m^3 should be kg/m^3. The mass density of ZrSeTe can not be as large as 117231 kg/m^3 (see Table 1).

Ans3: We thank the reviewer for this valuable comment. Indeed, the mass density of the ZrSeTe presented in Table 1 is just a typo (copy/paste from another data) and that value was not used in our calculation. We have corrected this typo in the revised manuscript. Besides, we have checked the spelling mistakes over the whole manuscript as suggested by the reviewer. For instance, we changed the Km/s to km/s and Kg/m^3 to kg/m^3 in Table 1.

Reviewer 2 Report

Comments and Suggestions for Authors

Marfoua and Hong present an ab-initio study of ZrSeTe layered Janus material and calculate in detail the thermoelectric transport properties. The study has two particularly strong points. Firstly, it explains in detail the relevant thermoelectric properties for active cooling applications. This is by far less well known in the thermoelectric community than the zT for power-generation. Second, on a more technical level, the authors make a great effort to calculate the electrtonic and phonon relaxation times. These are crucial for calculating the transport properties but since it is a difficult task, it is often skipped in DFT studies.

My main issue with the manuscript is that neither from the title, nor the abstract can the reader find out that this is a DFT study. Also, the fact that the relaxation times are calculated could be mentioned here.

Minor comments:

the way the electronic relaxation times are calculated is explained in great detail. However, I did not find the details for how the phonon relaxation times of Fig.4c are obtained.

In many figures the transport coefficients are presented at various temperatures between 270-300K, but they show no relevant temperature dependence, and this is commented on several times. What is the point? Why not show only the 300K results, and mention once that calculations down to 270K show no important changes? Also, why not calculate over a broader range of temperature, in particular for higher ones where zT improves?

The out-of-plane properties are crucially important in this study. My understanding is that these properties are heavily influenced by the flat band commented on (Fig. 1c Gamma-A). I assume that this part of the band structure is very sensitive to how the van der Waals interaction is handled in the DFT. Since this is usually delicate/controversial, it should be commented on in detail. 

The density values in Table 1 (last column) must be several orders of magnitude wrong. Hopefully this is just a typo and did not actually influence the property calculations. This should be checked.

Author Response

ANSWERS TO REVIEWERS

We thank the reviewers very much for reading our manuscript carefully and providing valuable comments and suggestions, which help us to enhance the quality of the work. That will greatly improve the manuscript and we have tried to do our best to respond to the points raised.

Reviewers have brought up worthy points and we appreciate the opportunity to clarify our research objectives and results. We have provided a detailed point-to-point response to address the following concerns. We have made necessary changes accordingly to their indications.

Our response follows.

Reviewer #2:

Comments and Suggestions for Authors

Marfoua and Hong present an ab-initio study of ZrSeTe layered Janus material and calculate in detail the thermoelectric transport properties. The study has two particularly strong points. Firstly, it explains in detail the relevant thermoelectric properties for active cooling applications. This is by far less well known in the thermoelectric community than the zT for power-generation. Second, on a more technical level, the authors make a great effort to calculate the electrtonic and phonon relaxation times. These are crucial for calculating the transport properties but since it is a difficult task, it is often skipped in DFT studies.

C1. My main issue with the manuscript is that neither from the title, nor the abstract can the reader find out that this is a DFT study. Also, the fact that the relaxation times are calculated could be mentioned here.

Ans1: We thank the reviewer for this valuable comment. Indeed, we acknowledge the importance of providing clear and upfront information about the study's approach to enhance the reader's understanding, as the reviewer mentioned. We revise the title to explicitly indicate the DFT (Density Functional Theory) approach. Also, we include a concise statement highlighting the utilization of DFT for the study in the abstract. Additionally, we explicitly describe the calculation of relaxation times to provide a more comprehensive overview of the used methodology. We appreciate the constructive form of the reviewer which aims to ensure that the readers promptly grasp the nature of the study, emphasizing the use of DFT and the calculation of relaxation times.

Minor comments:

C2. The way the electronic relaxation times are calculated is explained in great detail. However, I did not find the details for how the phonon relaxation times of Fig.4c are obtained.

Ans2: We thank the reviewer for this valuable comment and give us the opportunity to clarify this point. Indeed, we recognize the importance of providing comprehensive details about the calculation of phonon relaxation times, as was done for electronic relaxation times. Indeed, the electronic thermal conductivity (κe), based on the Boltzmann transport theory, can be written as 2

                                                (R1)

where σαβ(ε), fμ , T, e, ε, μ,  represent the conductivity tensor, the Fermi-Dirac distribution function, the temperature, the elementary charge, the electron energy state, and the chemical potential.

The single-mode relaxation time τλ was calculated from the phonon line width Γ of the phonon mode λ using the relation3

                                                                                     (R2)

where Γ can be written as

                                         (R3)

Here, the Φ-λλ’λ’’ is the phonon-phonon interaction strength among three phonons λ, λ’, and λ’’ and this can be calculated from the second and third-order force constant while Nλ’λ’’ (ω) is

                                            (R4)

With .

We included further details about the procedure for obtaining phonon relaxation times in the revised manuscript, as the reviewer suggested in

C3. In many figures the transport coefficients are presented at various temperatures between 270-300K, but they show no relevant temperature dependence, and this is commented on several times. What is the point? Why not show only the 300K results, and mention once that calculations down to 270K show no important changes? Also, why not calculate over a broader range of temperature, in particular for higher ones where zT improves?

Ans3: We thank the reviewer for this valuable comment and give us the opportunity to clarify this point. In this study, we address the thermoelectric power generation as well as the thermoelectric cooling effects. Indeed, thermoelectric cooling applications are typically evaluated within a narrow temperature range, commonly around room temperature, with a focus on small temperature differentials. This is inherently tied to the practical considerations of thermoelectric cooling applications. The upper limit for such applications is typically set by room temperature, and the cooling system is generally not intended to generate cooling below-freezing temperatures (~ 270 K or 260 K). Given these practical constraints and the specific application context, our selection of a small temperature interval (300 K to 270 K) aligns with the typical operational range and objectives of thermoelectric cooling systems in real applications. Note that the cooling performances are calculated based on the transport coefficients and these cooling performances are sensitive to small temperate differences (ΔT = 30 K) as can be seen in Fig. 6.

We sincerely appreciate the constructive feedback provided by the reviewer and hope that our explanation satisfactorily addresses the concerns raised

C3. The out-of-plane properties are crucially important in this study. My understanding is that these properties are heavily influenced by the flat band commented on (Fig. 1c Gamma-A). I assume that this part of the band structure is very sensitive to how the van der Waals interaction is handled in the DFT. Since this is usually delicate/controversial, it should be commented on in detail.

Ans3: We express our gratitude to the reviewer for their valuable comment. Indeed, the band dispersion analysis suggests that out-of-plane transport is likely dominated by the valence band (hole doping). This inference is particularly influenced by the presence of a very flat band in the conduction band minimum (electron doping), as illustrated in the flat band of Figure 1(c) (Gamma-A). The existence of such a flat band implies an exceptionally large effective mass for electron carriers, leading to notably poor mobility. In light of these considerations, we have exclusively focused on the hole-doped system in our study. This choice is based on the expectation that the hole-doped configuration would yield more favorable out-of-plane transport characteristics.

Furthermore, we agree with the reviewer's insight regarding the sensitivity of van der Waals interactions, especially in systems exhibiting vdW interactions along the out-of-plane direction since vdW interactions can be delicate and sometimes controversial in DFT calculations. To accurately account for these interactions, we have incorporated the Grimme DFT-D3 method4 in our calculations. This approach ensures a comprehensive treatment of van der Waals forces between layers, particularly in the context of the out-of-plane transport properties.

C4. The density values in Table 1 (last column) must be several orders of magnitude wrong. Hopefully this is just a typo and did not actually influence the property calculations. This should be checked.

Ans4:  We appreciate the reviewer's diligence in identifying a potential discrepancy in the density values presented in Table 1 regarding the mass density of the ZrSeTe material. Indeed, that was just a typo (copy/paste from another data) and that value was not used in our calculation. We promptly corrected the typos in the revised manuscript

Reviewer 3 Report

Comments and Suggestions for Authors

Here, the authors investigated thermoelectric performance of bulk ZrSeTe Janus structure by DFT calculation. It is found that a maximum ZT value of ~0.065 can be achieved in the out-of-plane direction at 300 K which is one order of magnitude larger than that  in the in-plane direction (ZT ~ 0.006). The in-plane direction shows a cooling performance of 13 W/m.K and a coefficient of performance (COP) of ~ 90 with a temperature difference (ΔT) of 30 K, while the out-of-plane direction has a cooling performance of 2.5 W/m.K and COP of ~2.5.

This is a comprehensive theoretical investigation of material performance. However, the current version can be hardly accepted for publication considering the controversial understanding to traditional thermoelectrics without clearly scientific clarification.

1. It is necessary to clearly state this is a theoretical calculation study in the Abstract, to avoid confusion from readers.

2. COP of thermoelectric cooler is strongly related with temperature difference. For this reason, for effective and accurate comparison, it is better mentioning this pre-condition before comparison of COP between different techniques.

3. In the relationship-analysis between COP of thermoelectrics and material ZT, Joule heating, thermal conductance, and Peltier effect have all been considered. On this basis, it is well known the thermoelectric COP is dominated by ZT, under the scenario of certain temperature difference. Thus, it is confusing the authors claimed high ZT and low COP along out-of-plane direction, simultaneously. Please check and think carefully. If this conclusion (controversial to traditional understanding) is what the authors wish to claim, at least, the scientific finding should be clarified and highlighted, which has been not yet.

4. ‘Thermoelectric performance’ is a general term, not a specific and accurate one, including cooling capacity, cooling density, COP, etc. Here, the author use it as a specific term which is quite misleading.

Author Response

ANSWERS TO REVIEWERS

We thank the reviewers very much for reading our manuscript carefully and providing valuable comments and suggestions, which help us to enhance the quality of the work. That will greatly improve the manuscript and we have tried to do our best to respond to the points raised.

Reviewers have brought up worthy points and we appreciate the opportunity to clarify our research objectives and results. We have provided a detailed point-to-point response to address the following concerns. We have made necessary changes accordingly to their indications.

Our response follows.

Reviewer #3:

Here, the authors investigated thermoelectric performance of bulk ZrSeTe Janus structure by DFT calculation. It is found that a maximum ZT value of ~0.065 can be achieved in the out-of-plane direction at 300 K which is one order of magnitude larger than that  in the in-plane direction (ZT ~ 0.006). The in-plane direction shows a cooling performance of 13 W/m.K and a coefficient of performance (COP) of ~ 90 with a temperature difference (ΔT) of 30 K, while the out-of-plane direction has a cooling performance of 2.5 W/m.K and COP of ~2.5.

This is a comprehensive theoretical investigation of material performance. However, the current version can be hardly accepted for publication considering the controversial understanding to traditional thermoelectrics without clearly scientific clarification.

C1. It is necessary to clearly state this is a theoretical calculation study in the Abstract, to avoid confusion from readers.

Ans1: We thank the reviewer for this valuable comment. Indeed, we acknowledge the importance of providing clear and upfront information about the study's approach to enhance the reader's understanding. Thus, we included a concise statement highlighting the utilization of DFT for the study in the abstract of the revised manuscript.

C2. COP of thermoelectric cooler is strongly related with temperature difference. For this reason, for effective and accurate comparison, it is better mentioning this pre-condition before comparison of COP between different techniques.

Ans2: We appreciate the reviewer's insightful comment, and we agree that the coefficient of performance (COP) of a thermoelectric cooler is indeed closely tied to the temperature difference (ΔT). Where the COP increases with increasing the ΔT as indicated in Eq (12) in the revised manuscript. We provided clear and explicit clarification of this factor in the revised manuscript, as the reviewer suggested.

C3. In the relationship-analysis between COP of thermoelectrics and material ZT, Joule heating, thermal conductance, and Peltier effect have all been considered. On this basis, it is well known the thermoelectric COP is dominated by ZT, under the scenario of certain temperature difference. Thus, it is confusing the authors claimed high ZT and low COP along out-of-plane direction, simultaneously. Please check and think carefully. If this conclusion (controversial to traditional understanding) is what the authors wish to claim, at least, the scientific finding should be clarified and highlighted, which has been not yet.

Ans3: We appreciate the valuable comment from the reviewer, allowing us to address the perceived contradiction regarding the claimed high ZT and low coefficient of performance (COP) along the out-of-plane direction.

It's essential to distinguish between refrigeration and active cooling modes in thermoelectric systems. Thermoelectric cooling comprises two modes: refrigeration mode and active cooling mode (Figure R1). In the refrigeration mode, heat is transferred from the cold side to the hot side, resulting in a cooling effect on the cold side. Conversely, the active cooling mode is employed in thermoelectric coolers for electronic cooling applications that it is used to remove heat generated by specific components or devices. This means that the heat extracted from the hot side to the cold side follows the natural conduction heat flux which is in the same direction as the Peltier current.

(b)

(a)

Figure R1  Two ways to cool. (a) A refrigerator transfers heat from a cold object to its warmer surroundings (left). (b) By contrast, an active cooling system helps heat move more efficiently in its natural direction, from a hot object to the cooler surroundings (right). A new active cooling system is optimized for this second purpose and may be useful for cooling hot components.

In contrast to the refrigeration mode, ZT is not the sole determinant in the active cooling mode, but also thermal conductivity is crucial for efficiency5-8. In this active cooling mode, the thermal conductivity should be high for high cooling efficiency.

The COP in the cooling mode can be expressed as6 

                                                                      (R6)

Based on this, larger thermal conductivity and consequently lower ZT are advantageous for higher COP in the active cooling system.

In our study, we focused on the active cooling mode of the layered ZrSeTe Janus system (not the refrigeration performances). The anisotropic electrical and thermal properties of this system, stemming from its layered structure, result in larger ZT along the out-of-plane direction and higher thermal conductivity in the in-plane direction. This unique configuration allows the ZrSeTe Janus system to demonstrate superior thermoelectric power generation in the out-of-plane direction and enhanced thermoelectric cooling performance in the in-plane direction.

We hope that the reviewer finds out that this clarification aligns our conclusions with our results, and we appreciate the reviewer's engagement in ensuring clarity in our study.

C4. ‘Thermoelectric performance’ is a general term, not a specific and accurate one, including cooling capacity, cooling density, COP, etc. Here, the author use it as a specific term which is quite misleading.

Ans4: We appreciate the reviewer's observation regarding the term 'Thermoelectric performance' and acknowledge the need for precision in scientific terminology. To enhance clarity and precision, we used more specific and accurate terms when discussing different aspects of thermoelectric performance in the revised manuscript, as the reviewer suggested. We thank the reviewer for bringing this to our attention.

References

  1. Markov, M., Rezaei, S. E., Sadeghi, S. N., Esfarjani, K. & Zebarjadi, M. Thermoelectric properties of semimetals. Phys. Rev. Mater. 3, 095401 (2019).
  2. Madsen, G. K., Carrete, J. & Verstraete, M. J. BoltzTraP2, a program for interpolating band structures and calculating semi-classical transport coefficients. Comput. Phys. Commun. 231, 140-145 (2018).
  3. Togo, A., Chaput, L. & Tanaka, I. Distributions of phonon lifetimes in Brillouin zones. Phys. Rev. B 91, 094306 (2015).
  4. Grimme, S., Antony, J., Ehrlich, S. & Krieg, H. A consistent and accurate ab initio parametrization of density functional dispersion correction (DFT-D) for the 94 elements H-Pu. The Journal of chemical physics 132, 154104 (2010).
  5. Adams, M., Verosky, M., Zebarjadi, M. & Heremans, J. Active peltier coolers based on correlated and magnon-drag metals. Phys. Rev. Appl 11, 054008 (2019).
  6. Zebarjadi, M. Electronic cooling using thermoelectric devices. Appl. Phys. Lett. 106 (2015).
  7. Nimmagadda, L. A., Mahmud, R. & Sinha, S. Materials and devices for on-chip and off-chip peltier cooling: A review. IEEE Transactions on Components, Packaging and Manufacturing Technology 11, 1267-1281 (2021).
  8. Medrano Sandonas, L. et al. Anisotropic thermoelectric response in two-dimensional puckered structures. The Journal of Physical Chemistry C 120, 18841-18849 (2016).

Round 2

Reviewer 3 Report

Comments and Suggestions for Authors

I can not agree with the Authors' response, and do not think this study is suitable for publication.

Main concern is same as my previous comment 3:

After reading through the authors' response, I think the main problem is the understanding of heat conduction. Based on my understanding, regardless of active/passive cooling, the direction heat conduction is always from hot to cold sides, which looks like different from the authors' understanding. This might be the reason why their COP-ZT relationship is different from typical understanding. Additionally, COP is not solely determined by ZT and temperature, it is also strongly related with current. I assume the authors are trying to say maximum COP here. Please pay attention to accurate and scientific expression.

Author Response

ANSWERS TO REVIEWERS

We thank the reviewer very much for reading our manuscript carefully and providing valuable comments and suggestions, which help us to enhance the quality of the work. That will greatly improve the manuscript and we have tried to do our best to respond to the points raised.

The reviewer has brought up worthy points and we appreciate the opportunity to clarify our research objectives and results. We have provided a detailed response to address the following concerns. We have made the necessary changes according to their indications.

Our response follows

Reviewer #3:

  1. After reading through the authors' response, I think the main problem is the understanding of heat conduction. Based on my understanding, regardless of active/passive cooling, the direction heat conduction is always from hot to cold sides, which looks like different from the authors' understanding. This might be the reason why their COP-ZT relationship is different from typical understanding. Additionally, COP is not solely determined by ZT and temperature, it is also strongly related with current. I assume the authors are trying to say maximum COP here. Please pay attention to accurate and scientific expression.

Ans: We appreciate the reviewer's continued engagement with our study and their insightful comments. We are afraid there is still a misleading about our concept of heat conduction in thermoelectric cooling systems. We would like to reiterate that our perspective of heat conduction aligns with the conventional understanding.

Indeed, the thermal conduction is definitely flowing from hot to cold sides (the natural flow) regardless of the thermoelectric refrigeration or cooling modes. In Figure R1 (below), we illustrate the difference between refrigeration and cooling modes in thermoelectric systems.

Figure R1  Two ways in thermoelectric cooling. (a) A refrigerator transfers heat from a cold object to its warmer surroundings (left). (b) By contrast, an active cooling system helps heat move more efficiently in its natural direction, from a hot object to the cooler surroundings (right).

It's essential to distinguish between refrigeration and cooling modes in thermoelectric systems. In the refrigeration mode, the Peltier current (αI) is going against the natural thermal conductance (K) (Fig. R1(a)). Hence, a much larger Peltier current is needed to overcome the natural flow of the thermal conductance in order to transfer the heat (QC) from the cold side to the hot side. This results in a cooling effect on the cold side. However, in the active cooling mode (Fig. R1(b)), one needs to reverse the Peltier current (αI) to follow along with the natural thermal conductance (K) direction. This means that both Peltier current and thermal conductance contribute to extracting the heat (QH) from the hot side to the cold side.

As a result, the two modes have different corresponding COP; COP = QC/W for the refrigeration mode and COP = QH/W for the cooling mode. The COP of a TE refrigerator mode is given by1

   =                                                                   (R1)

As can be seen, the COP in the refrigeration mode is increasing as a function of Z and larger temperature difference.

However, the COP in the thermoelectric cooling mode is expressed as1 

   =                                                                   (R2)

Unlike the refrigeration mode, the thermoelectric cooling performance is proportional to thermal conductivity (k) and temperature difference. However, as the Z increases the COP decreases.

Regarding the influence of the current (I) in the COP, we acknowledge the importance of this factor and agree with the reviewer on this point. In our study, we focus on density-based quantities where we consider the conductivity in estimating ZT. Note that the current (I) is s implicitly considered in the conductance that can be derived from conductivity and sample size.

Indeed, we are referring to the maximum COP in our study. We have duly amended our manuscript to explicitly refer to the maximum COP as COPmax, as suggested by the reviewer, ensuring accuracy in our terminology.

In the end, we hope that our clarification on the two modes of cooling is helpful. One involves a thermoelectric refrigerator transferring heat from a cold object to its warmer surroundings. In contrast, the thermoelectric active cooling system facilitates more efficient heat movement in its natural direction, from a hot object to cooler surroundings.

We appreciate the reviewer's guidance, and these clarifications will be seamlessly incorporated into the revised manuscript.

References

  1. Zebarjadi, M. Electronic cooling using thermoelectric devices. Appl. Phys. Lett. 106 (2015).

Round 3

Reviewer 3 Report

Comments and Suggestions for Authors

I appreciate the authors response.

The schematic diagram Figure R1 is still confusing. It looks like the difference between refrigerator and active cooling modes, are the semiconductor type. In other words, p/n-type semiconductors have Seebeck with different symbols, indicating different current direction during Peltier cooling here. However, in typical Peltier cooler, we are considering the static state, where the final state of a Peltier cooler should lead to a heat conduction direction (hot to cold side) against the current direction in n-type materials and along the current direction in n-type materials. It should be noted this typical understanding is under static assumption, where the hot and cold ends are thermally stable. This means at each end, the overall heat absorption equals to the heat dissipation.

After reading the reference and the response, my understanding is, here the the authors are talking about dynamic process, where the heat dissipation and absorption at cold/hot ends are not necessary under equilibrium status. For this case, the expression might be correct, but this COP will definitely be different from traditional understanding, which need to be specified and defined differently.

Author Response

ANSWERS TO REVIEWERS

We thank the reviewer very much for reading our manuscript carefully and providing valuable comments and suggestions, which help us to enhance the quality of the work. That will greatly improve the manuscript and we have tried to do our best to respond to the points raised.

The reviewer has brought up worthy points and we appreciate the opportunity to clarify our research objectives and results. We have provided a detailed response to address the following concerns. We have made the necessary changes according to their indications.

Our response follows

Reviewer #3:

  1. The schematic diagram Figure R1 is still confusing. It looks like the difference between refrigerator and active cooling modes, are the semiconductor type. In other words, p/n-type semiconductors have Seebeck with different symbols, indicating different current direction during Peltier cooling here. However, in typical Peltier cooler, we are considering the static state, where the final state of a Peltier cooler should lead to a heat conduction direction (hot to cold side) against the current direction in n-type materials and along the current direction in n-type materials. It should be noted this typical understanding is under static assumption, where the hot and cold ends are thermally stable. This means at each end, the overall heat absorption equals to the heat dissipation.

After reading the reference and the response, my understanding is, here the the authors are talking about dynamic process, where the heat dissipation and absorption at cold/hot ends are not necessary under equilibrium status. For this case, the expression might be correct, but this COP will definitely be different from traditional understanding, which need to be specified and defined differently.

Ans:

We appreciate the reviewer's diligence in scrutinizing our work and his insightful comments. We acknowledge that this dynamic approach might lead to a different interpretation of the coefficient of performance (COP) compared to the traditional understanding, as correctly identified by the reviewer. We will take this valuable feedback into account and ensure that the dynamic nature of the process and its implications on COP are clearly defined and specified in the revised manuscript.

We appreciate the reviewer's careful consideration and these clarifications to the refine of our study. Thank you for your constructive feedback.
